# Effects of high-intensity interval training on selected indicators of physical fitness among male team-sport athletes: A systematic review and meta-analysis

**Yandong Yuan**[1,2], **Kim Geok Soh**[1]*, **Fengmeng Qi**[1,2], **Marrium Bashir**[1], **Ningxia Zhao**[3]

**1** Department of Sports Studies, Faculty of Educational Studies, Universiti Putra Malaysia, Selangor, Malaysia, **2** School of Physical Education, Henan Polytechnic University, Jiaozuo, China, **3** School of Applied Foreign Language, Henan Industry and Trade Vocational College, Zhengzhou, China

* kims@upm.edu.my

## Abstract

**Data Availability Statement:** All relevant data is within the manuscript and its Supporting Information files.

### Background

Superior physical fitness and performance are essential in male team sports. Among a myriad of training methodologies, high-intensity interval training (HIIT) has gained popularity owing to its unparalleled efficiency and effectiveness. Previous studies have established that HIIT is a proven and effective approach for enhancing various physiological performance outcomes, particularly oxygen consumption capacity, in individual sports. Despite potential differences in training practices between male and female athletes, HIIT is recognized as an anaerobic training approach for team-sport athletes. This systematic review aimed to comprehensively and innovatively analyze the existing literature to examine the effectiveness of HIIT on oxygen consumption performance among male team-sport athletes.

### Methods

A comprehensive literature search was conducted in accordance with the Preferred Reporting Items for Systematic Reviews and Meta-Analyses (PRISMA) guidelines across the PubMed, SCOPUS, Web of Science, and SPORTDiscus databases until December 31, 2023. The inclusion criteria for this review encompassed research articles published in peer-reviewed journals that specifically focused on the impact of HIIT on the oxygen consumption performance of male players engaged in team sports. The study population exclusively consisted of male participants. The collected data included study characteristics, participant demographics, intervention details, and outcomes. Methodological quality assessment was performed using standardized criteria. The effect sizes (ESs) were calculated, and a meta-analysis was conducted using a random-effects model.

**Funding:** The author(s) received no specific funding for this work.

**Competing interests:** The authors have declared that no competing interests exist.

## Results

The literature search yielded 13 eligible studies encompassing 286 athletes aged 14–26 years. The meta-analysis showed statistically significant enhancements in maximal oxygen uptake (VO2max) in six studies (ES, 0.19−0.74; $p < 0.005$), Yo-Yo Intermittent Recovery Test (YYIRT) performance in six studies (ES, 0.20−2.07; $p = 0.009$), repeated-sprint ability total time ($RSA_{total}$) in five studies (ES, 0.18−1.33; $p < 0.001$), and the best and average times for repeated-sprint ability ($RSA_{best}$ and $RSA_{mean}$, respectively) in four studies (ES, 0.47−1.50; $p < 0.001$). However, two studies did not report any significant differences in the outcomes of the Velocity in 30–15 Intermittent Fitness Test (VIFT) between the experimental and control groups (ES, −0.08 and −0.27; $p = 0.87$ and 0.443, respectively). Moreover, one study did not report any significant differences in the maximal aerobic speed (MAS) (ES, 0.41, $p = 0.403$).

## Conclusions

HIIT significantly improved VO2max, YYIRT, and RSA; however, it did not appear to enhance VIFT and MAS performance, irrespective of age or competition level. These findings indicate that HIIT could serve as a valuable method for improving oxygen consumption performance (VO2max, YYIRT, and RSA) in male team-sport athletes, offering a time-efficient alternative to the traditional training methods. Further research is warranted to investigate its impact on performance outcomes in competitive settings and identify optimal HIIT protocols tailored to specific team sports.

## Introduction

Physical fitness is a fundamental component of competitive sports, encompassing not only muscular strength and flexibility but also cardiorespiratory fitness, aerobic endurance, and repeated-sprint ability (RSA) [1, 2]. These elements critically influence athletes' performance and stability [3, 4]. Aerobic capacity and endurance are pivotal constituents of physical fitness, with aerobic capacity, specifically maximum oxygen uptake (VO2max), reflecting the body's ability to consume oxygen during exercise and serving as a crucial indicator of an athlete's endurance [5]. Aerobic endurance determines an athlete's proficiency during extended periods of low- to moderate-intensity exercise, directly affecting their physical resilience and recovery rates [6].

Another significant factor is RSA, which is particularly vital in team sports such as football, basketball, and rugby [7]. These sports require athletes to repeatedly engage in high-intensity sprints within short time frames while swiftly recovering between sprints [8]. A heightened RSA enhances performance during critical moments and prolongs a team's overall efficiency throughout the game, thereby improving competitiveness [7]. Additionally, in team ball sports, RSA influences the ability to compete for ball possession through numerous rapid sprints [9].

Therefore, an exceptional level of physical fitness not only enhances individual performance, but also fortifies team dynamics, underscoring the importance of employing scientific training methodologies [10]. Efficient and practical training methods, such as sustained long-distance running, repetitive sprint training, and high-intensity interval training (HIIT), have demonstrated significant potential in enhancing athletes' performance [11]. These training

modalities collectively improve the various aspects of physical fitness, ensuring athletes are well-prepared to meet the demands of their respective sports and contributing to their overall success in competition.

In team sports, the outcome of a game typically hinges on the collective physical performance of athletes [12, 13]. Team sports often require athletes to engage in short, highly repetitive, or near-maximal effort tasks, such as accelerated running, sprinting, sharp stopping and turning, vertical jumping, and other high-intensity activities that are frequently interspersed with brief recovery periods [14]. For example, in football, an athlete's aerobic capacity and endurance play a crucial role in determining their effective involvement in attacking and defending throughout the game. Consequently, scientific training methods, including aerobic and anaerobic training, are essential.

HIIT, a method combining aerobic and anaerobic training, has garnered widespread attention for its ability to significantly improve athletes' fitness levels [15]. Through systematic training, athletes can significantly enhance their fitness levels and contribute to better team performance in competitions [16]. HIIT is characterized by alternating periods of vigorous exercise and brief recovery intervals, involving short bursts of high-intensity exercise followed by relaxation or lower-intensity exercise [17]. It encompasses a variety of activities that can be customized based on intensity and duration, rather than being limited to a specific program [18]. The HIIT method offers significant advantages compared to traditional training methods [19–22]. This form of training can be accomplished within a relatively concise timeframe using minimal or no equipment [21]. Moreover, it facilitates faster physiological adaptation of the body than traditional endurance training methods [23]. This reduced timeframe and significant fitness improvements have contributed to HIIT's popularity [24].

Studies indicate the efficacy of HIIT in enhancing youth fitness and its widespread adoption in daily fitness routines and team-sport training [25, 26]. Nine systematic reviews and meta-analyses investigating the impact of HIIT on fitness outcomes in both adults and teenagers consistently concluded that HIIT is safe and effective [25, 27–34]. HIIT could serve as a more viable option for non-professional sports teams facing time and facility constraints, ensuring the effectiveness of their training regimen [25].

A previous systematic review of women's team sports revealed that HIIT significantly improved both VO2max and RSA [16]. However, male and female athletes differ significantly, making it difficult to generalize the results of studies across sexes [35]. Additionally, previous systematic reviews on the use of HIIT in team sports have primarily focused on male footballers [32, 36], basketball players [37, 38], and handball players [39]. However, the oxygen consumption performances of different male team-sport athletes have not yet been compared. This systematic review aimed to provide a comprehensive overview of the impact of HIIT on oxygen consumption performance among male athletes participating in various team sports across different disciplines.

## Materials and methods

### Protocol and registration

This review was conducted in accordance with the guidelines outlined by the Preferred Reporting Items for Systematic Reviews and Meta-Analyses [40], encompassing data selection, collection, and analysis. This research project was duly recorded on the International Platform for Registered Systematic Reviews and Meta-Analyses Protocols (http://inplasy.com/;reg.no.: INPLASY202310028;DOI:10.37766/inplasy2023.1.0028).

## Eligibility criteria

The selection criteria for the studies in this review were determined based on the Population, Intervention, Comparison, Outcomes, and Study (PICOS) framework.

- Population: This study included male team-sport athletes in good health, regardless of age and competition level, with no specific restrictions.

- Interventions: Exercise interventions should have lasted 4–9 weeks. The intervention program consisted of alternating periods of interval running and repetitive sprint training, incorporating interspersed active or passive recovery periods. The exercises primarily involved sprinting without assistive devices. To minimize confounding effects, this study excluded a hybrid training modality that combined HIIT with plyometric or functional modalities. The HIIT and control groups were subjected to identical training regimens and durations.

- Comparison: The control group should have undergone small-sided games (SSGs) or routine technical and tactical training, excluding HIIT.

- Outcomes: The study outcomes should have encompassed the impact of at least one HIIT on participants' aerobic capacity or RSA. Aerobic capacity was assessed by measuring VO2max, the Yo-Yo Intermittent Recovery Test (YYIRT), and Velocity in 30–15 Intermittent Fitness Test (VIFT), while repetitive sprinting capacity was evaluated using a short shuttle run. To meet the inclusion criteria, the studies should have reported pre- and post-test values for treatment effects or relevant test metrics. Studies focusing solely on sport-specific technical skills were excluded.

- Study design: This review included only randomized controlled trials that satisfied the predetermined inclusion criteria.

## Search strategy and selection process

A comprehensive search was conducted using four well-known scientific databases: PubMed, SCOPUS, Web of Science, and SPORT Discus. The search included studies published until December 31, 2023. In each of the four databases, the searches were performed by title, using a pre-defined combination of keywords: ("high-intensity interval training" OR "high-intensity intermittent training" OR "HIIT") AND ("male team athletes" OR "handball" OR "basketball" OR "soccer" OR "football") AND ("physical performance" OR "maximal oxygen uptake" OR "VO2max" OR "aerobic capacity" OR "repeated sprint ability"). Simultaneously, the literature search involved incorporating citations, including additional studies from the reference list, and a retrospective analysis of the bibliographies included in previous review articles. All relevant topics were manually searched to identify all possible articles for inclusion. In addition, screening was conducted using Google Scholar (Alphabet Inc., Mountain View, CA, USA) to identify noteworthy recent studies.

The study selection process comprised four steps (Fig 1). After conducting a comprehensive literature search, relevant studies were integrated into the reference management software to eliminate duplicate entries. Initially, an experienced librarian from Universiti Putra Malaysia provided guidance and assistance in formulating a systematic search strategy. The initial screening phase was conducted by two separate reviewers (YY and FQ), who evaluated the titles and abstracts of the studies. The exclusion of irrelevant materials was performed according to pre-established criteria. To ensure a thorough examination, this systematic review and

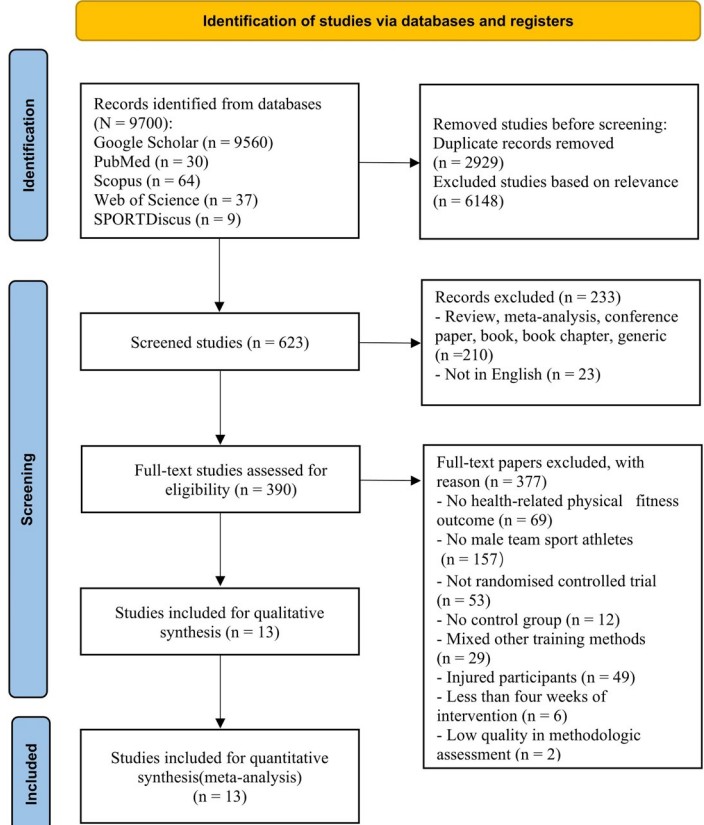

**Fig 1. PRISMA flow diagram.**

meta-analysis exclusively considered articles written in English for potential inclusion. Studies with incomplete text were excluded. If there were differences in opinion among the reviewers, a third reviewer (KGS) was consulted to reach an agreement. After eliminating irrelevant content from the database based on the pre-established exclusion and inclusion criteria, the remaining titles and abstracts were assessed.

## Data extraction

After completing the literature search, data from eligible studies were extracted using Microsoft Word spreadsheets (Microsoft Corp., Redmond, WA, USA) based on the PICOS index criteria (Table 1). This included information such as (1) author and publication year; (2) country; (3) athlete type and age; (4) sample size and group; (5) sports level and experience; (6) duration and session; (7) experimental comparison; and (8) research outcomes. The primary author was responsible for data extraction and entry. The process was subsequently evaluated by another author assisted by the corresponding author.

## Study quality assessment

Two independent evaluators (YY and FQ) used the PEDro scale to evaluate the potential bias of studies that met the predetermined inclusion criteria. The results were cross-checked by a third evaluator (MB), leading to a unanimous consensus. The PEDro scale, which can be accessed online at https://pedro.org.au/wp-content/uploads/PEDro.scale.pdf, is an extremely

**Table 1. Qualitative analysis of the included studies.**

| Authors | Country | Participants | | | Interventions | Comparison | | Outcomes | |
|---|---|---|---|---|---|---|---|---|---|
| | | Age (years)/ sports | Number/ groups | Level/ Experience (years) | | | | EG | CG |
| Arslan et al., 2022 [41] | Turkey | EG: 14.6 ± 0.5 CG: 14.4 ± 0.5 Basketball | N = 32 EG = 16 CG = 16 | Regional/ ≥ 3 | Total weeks: 6 Session (t/w): 3 Duration (min): 10–18 | HIIT | 2 × (6–9 min of 15"–15" intermittent runs) at 90–95% VIFT; 15 s/2 min recovery | VO2max ↑* YYIRT1 ↑* VIFT ↑ RSA$_{total}$ ↑* | VO2max ↑ YYIRT1 ↑ VIFT ↑ RSA$_{total}$ ↑ |
| | | | | | | SSG | 2 (2 × 2.30–4 min 2 vs. 2 games) at > 85% HR$_{max}$; 2 min recovery | | |
| Arslan et al., 2020 [42] | Turkey | EG: 14.1 ± 0.6 CG: 14.4 ± 0.5 Soccer | N = 20 EG = 10 CG = 10 | Academy / EG: 3.3 ± 0.3 CG: 3.4 ± 0.3 | Total weeks: 5 Session (t/w): 2 Duration (min): 12–20 | HIIT | 2 × (6–10 min 15"–15" continuous runs) at 90–95% VIFT; 15 s/2 min relax | VO2max ↑* YYIRT1 ↑* RSA$_{total}$ ↑* | VO2max ↑ YYIRT1 ↑ RSA$_{total}$ ↑ |
| | | | | | | SSG | 2× (2 × 2'30" –4'30") 2–a– side games; 2 min relax | | |
| Boraczyński et al., 2023 [43] | Poland | EG: 25.6 ± 3.98 CG: 24.3 ± 5.16 Soccer | N = 25 EG = 13 CG = 12 | National/ EG: 15.4 ± 1.46 CG: 17.3 ± 2.38 | Total weeks: 6 Session (t/w): 2–3 Duration (min): 20 | SIT | Phase 1: (2·wk−1): 1 set, 10 × 45 s, 1 min active recovery Phase 2: (3·wk−1): 2 sets, 10 × 30 s, 30 s active recovery | VO2max → | VO2max ↑ |
| | | | | | | SSG | Phase 1: (2·wk−1): 5 × 3 min, 3 min active recovery Phase 2: (3·wk−1): 4 × 4 min, 3 min active recovery | | |
| Delextrat et al., 2018 [44] | United Kingdom | EG: 14.4 ± 0.5 CG: 14.1 ± 0.6 Basketball | N = 17 EG = 9 CG = 8 | National/ 5.2 ± 3.6 | Total weeks: 6 Session(t/w): 2 Duration(min): NR | HIIT | 2 × (8–13 min intermittent running at 95% of VIFT); 15 s rest | VIFT ↑ | VIFT ↑ |
| | | | | | | SSG | 2 × (2–3 × 3–5 min 2 vs. 2 games) all out; 2 min recovery | | |
| Eniseler et al., 2017 [45] | Turkey | 16.9 ± 1.1 Soccer | N = 19 EG = 9 CG = 10 | National/ EG: 4.44 ± 0.88 CG: 5.6 ± 1.17 | Total weeks: 6 Session (t/w): 2 Duration (min): 60–90 | RST | 3 × (6 × 40 m) sprint running 20 s / 4 min passive recovery | YYIRT1 ↑ RSA$_{best}$ ↓ RSA$_{mean}$ → | YYIRT1 ↑ RSA$_{best}$ ↓ RSA$_{mean}$ ↓ |
| | | | | | | SSG | 4 × 3 min workout; 4 min passive recovery between sets, at 90–95% HR$_{max}$ | | |
| Gantois et al., 2019 [46] | Brazil | 21.2 ± 2.3 Basketball | N = 17 EG = 9 CG = 8 | College/ > 3 | Total weeks: 6 Session (t/w): 2 Duration (min): NR | RST | 2–3 × (6 × 30 m) at 90–95% HR$_{max}$; 1–3 min/20 s rest | VO2max ↑ RSA$_{total}$ ↑ RSA$_{best}$ ↑ RSA$_{mean}$ ↑ | VO2max ↓ RSA$_{total}$ ↓ RSA$_{best}$ ↑ RSA$_{mean}$ ↓ |
| | | | | | | CG | Physical–technical exercises | | |
| Hermassi et al., 2018 [47] | Tunisia | EG: 17.1 ± 0.3 CG: 17.3 ± 0.5 Handball | N = 30 EG = 15 CG = 15 | National/ ≥ 8 | Total weeks: 7 Session (t/w): 2 Duration (min): NR | SIT | 1–4 × (5–10 × 10–20 s) at 110–130% of MAS; 10–20 s rest | YYIRT1 ↑ RSA$_{total}$ ↑ RSA$_{best}$ ↑ | YYIRT1 ↓ RSA$_{total}$ → RSA$_{best}$ ↑ |
| | | | | | | CG | 5 × 90 min skill drills, offensive and defensive tactics | | |
| Kavaliauskas et al., 2017 [48] | United Kingdom | EG: 22 ± 8 CG: 23 ± 7 Soccer | N = 14 EG = 7 CG = 7 | National/ ≥ 5 | Total weeks: 6 Session (t/w): 2 Duration (min): NR | UST | Uphill sprint training all–out | YYIRT1 ↑ | YYIRT1 → |
| | | | | | | CG | Maintain their current training | | |

*(Continued)*

**Table 1.** (Continued)

| Authors | Country | Participants | | | Interventions | Comparison | | Outcomes | |
|---|---|---|---|---|---|---|---|---|---|
| | | Age (years)/ sports | Number/ groups | Level/ Experience (years) | | | | EG | CG |
| Kumari et al., 2023 [49] | India | EG: 21.4 ± 2.6 CG: 21.9 ± 2.4 Basketball | N = 40 EG = 20 CG = 20 | National/ NR | Total weeks: 5 Session (t/w): 2 Duration (min): NR | HIIT | Session A: 4 × 4 min intervals; 3 min rest Session B: 2 sets 15 × 30 s, 15 s/3 min rest | VO2max ↑ | VO2max → |
| | | | | | | CG: | Regular fitness training | | |
| Lacono et al., 2015 [50] | Israel | 25.6 ± 6.5 Handball | N = 18 EG = 9 CG = 9 | National/ ≥ 5 | Total weeks: 8 Session (t/w): 2 Duration (min): NR | HIIT | 15 s sprint running at 90% of YYIRT1 final speed 15 s passive recovery | YYIRT1 ↑ | YYIRT1 ↑ |
| | | | | | | SSG | 5 bouts 2' 25" continuous handball small–sided games 1 min passive recovery | | |
| Los Arcos et al., 2015 [51] | Israel | EG: 15.8 ± 0.5 CG: 15.1 ± 0.7 Soccer | N = 15 EG = 8 CG = 7 | National/ 8.5 | Total weeks: 6 Session (t/w): 2 Duration (min): NR | IT | 3 bouts × 4 min running at 90–95% HR$_{max}$; 3 min relax (50–60% HR$_{max}$) | MAS ↑ | MAS → |
| | | | | | | SSG | 3 × 4 min of (3 vs. 3) or (4 vs. 4) or (4 + Goalie) vs. (4 + Goalie); 3 min rest | | |
| Salazar-Martínez et al., 2023 [52] | Colombia | EG: 19.5 ± 4.25 CG: 19.00 ± 2.00 Soccer | N = 23 EG = 12 CG = 12 | National/ EG: 12.58±1.93 CG: 9.82 ± 1.54 | Total weeks: 9 Session (t/w): 2 Duration (min): 20–40 | HIIT | Interval running (IT): 15"/ 15", 20"/20", 10"/20", and 30"/ 30", at > 90% HR$_{max}$ | RSA$_{total}$ ↑ RSA$_{best}$ ↑ RSA$_{mean}$ → | RSA$_{total}$ ↑* RSA$_{best}$ ↑ RSA$_{mean}$ ↑ |
| | | | | | | SSG | 2 vs. 2 to 6 vs. 6 formats, at 85% of the maximal aerobic speed | | |
| Wells et al., 2014 [53] | United Kingdom | 21.3 ± 2.1 Soccer | N = 16 EG = 8 CG = 8 | National/ full–time professionals for at least 2 years | Total weeks: 6 Session (t/w): 3 Duration (min): NR | SIT | 60 s, 35 s, and 10 s repeat intermittent running at ≥ 95% of HR$_{max}$ | VO2max ↑ YYIRT2 ↑ | VO2max → YYIRT2 → |
| | | | | | | CG | Normal training | | |

In this study, we conducted a controlled HIIT intervention on male team-sport athletes. ↑, strong positive effect; ↓, strong negative effect; ↔, no strong effect; NR, not reported; VO2max, maximal oxygen uptake; YYIRT1 or 2, Yo-Yo Intermittent Recovery test level 1 or 2; RSA, repeated-sprint ability; VIFT, Velocity in 30–15 Intermittent Fitness Test; RSA$_{total}$, total sprint time; RSA$_{best}$, best sprint time; RSA$_{mean}$, average sprint time; * strong difference between groups; MAS, maximal aerobic speed; HIIT, high-intensity interval training; SSG, small-side game; RST, repeated sprint training; SIT, short interval training; IT, interval training; UST, uphill sprint training; HR$_{max}$, maximal heart rate; EG, experimental group; CG, control group; t/w, times/week.

reliable tool primarily designed for assessing the quality of randomized controlled trials available on PEDro. This scale comprises 11 components aimed at conducting a comprehensive evaluation of methodological rigor in randomized controlled trials [54]. The PEDro scale comprises 11 questions, with each criterion receiving one point if met. No points are awarded for criteria that are not fulfilled. The first item on the scale does not contribute to the overall score; hence, a score of 10 signifies exceptional study quality, while scores ranging from 0–3, 4–5, 6–8, and 9–10 indicate poor, fair, good, and excellent quality, respectively [55, 56]. Furthermore, the inter-rater reliability of the PEDro score for clinical trials assessing physiotherapy-related interventions has been reported to range from "fair" to "excellent" (intra-class correlation coefficient [ICC] = 0.53 to 0.91) in previous investigations [54, 57, 58]. Thus, this approach has been used in previous systematic reviews to evaluate the quality of research [59–61]. Overall, 13 articles were analyzed, with three receiving a fair evaluation and the remaining 10 earning a good quality rating. These assessments indicate that the selected articles

demonstrated exceptional quality, validating the rigorous scientific approach undertaken during the selection process (Table 2).

## Statistical analysis

A comprehensive meta-analysis software package (version 3) was used for data analysis [63]. The effect sizes (ESs) were calculated using the mean and standard deviation of the performance measures obtained before and after the intervention. To maintain consistency, we standardized the data using post-intervention measurements from a comparable performance measure [64]. During the data analysis process, if there were missing data, we assumed an ES of 0.7 and conducted the relevant meta-analysis based on this assumption, which is supported by a previous study [65]. Adopting this approach ensures the completeness and consistency of the analysis, while providing a conservative estimate. An analysis of post-test data, the median and quartiles, and a similar baseline for the pre-test data was conducted in one of the included studies [52]. The classification of ESs was as follows: insignificant ($< 0.2$), minor (0.20–0.6), moderate (0.6–1.2), significant (1.2–2.0), highly significant (2.0–4.0), and exceptionally significant ($> 4.0$) [66].

$I^2$ statistics were used to evaluate the variability among the included studies. Values less than 50% indicate a minimal degree of heterogeneity, values ranging from 50% to 75% suggest moderate levels, and values exceeding 75% indicate substantial heterogeneity [67]. The statistical significance of the heterogeneity was evaluated using the chi-square test, with a p-value $< 0.05$ indicating significance. In the absence of statistical heterogeneity ($p > 0.05$ in the chi-square statistics), a fixed-effect model and 95% confidence intervals (CIs) were employed for conducting a meta-analysis [68]. However, when statistical heterogeneity was detected, a more conservative approach was adopted, utilizing a random-effects model and calculating the corresponding 95% CIs [68].

**Table 2. Assessment of bias in individual studies.**

| Study | No. 1 | No. 2 | No. 3 | No. 4 | No. 5 | No. 6 | No. 7 | No. 8 | No. 9 | No. 10 | No. 11 | Score* | Quality rating levels |
|---|---|---|---|---|---|---|---|---|---|---|---|---|---|
| Arslan et al., 2022 [41] | 0 | 1 | 0 | 1 | 0 | 0 | 0 | 1 | 1 | 1 | 1 | 6 | Good |
| Arslan et al., 2020 [42] | 0 | 1 | 0 | 1 | 0 | 0 | 0 | 1 | 1 | 1 | 1 | 6 | Good |
| Boraczyński et al., 2023 [43] | 1 | 1 | 0 | 1 | 0 | 0 | 0 | 1 | 1 | 1 | 1 | 6 | Good |
| Delextrat et al., 2018 [44] | 1 | 1 | 0 | 1 | 0 | 0 | 0 | 1 | 1 | 1 | 1 | 6 | Good |
| Eniseler et al., 2017 [45] | 0 | 1 | 0 | 0 | 0 | 0 | 0 | 1 | 1 | 1 | 1 | 5 | Fair |
| Gantois et al., 2019 [46] | 1 | 1 | 0 | 1 | 0 | 0 | 0 | 1 | 0 | 1 | 1 | 5 | Fair |
| Hermassi et al., 2018 [47] | 1 | 1 | 0 | 1 | 0 | 0 | 0 | 1 | 1 | 1 | 1 | 6 | Good |
| Kavaliauskas et al., 2017 [48] | 0 | 1 | 0 | 1 | 0 | 0 | 0 | 1 | 1 | 1 | 1 | 6 | Good |
| Kumari et al., 2023 [49] | 1 | 1 | 0 | 0 | 0 | 0 | 0 | 1 | 1 | 1 | 1 | 5 | Fair |
| Lacono et al., 2015 [50] | 1 | 1 | 0 | 1 | 0 | 0 | 0 | 1 | 1 | 1 | 1 | 6 | Good |
| Los Arcos et al., 2015) [51] | 1 | 1 | 0 | 1 | 1 | 1 | 0 | 1 | 1 | 1 | 1 | 8 | Good |
| Salazar-Martínez et al., 2023 [52] | 1 | 1 | 1 | 1 | 0 | 1 | 0 | 1 | 1 | 1 | 1 | 8 | Good |
| Wells et al., 2014 [53] | 0 | 1 | 0 | 1 | 0 | 0 | 0 | 1 | 1 | 1 | 1 | 6 | Good |

**Note 1:** No. 1) Inclusion criteria and source; No. 2, random allocation; No. 3, allocation concealment; No. 4, baseline comparability; No. 5, blind subjects; No. 6, blind therapists; No. 7, blind assessors; No. 7, adequate follow-up; No. 9, intention-to-treat analysis; No. 10, between-group comparison; No. 11, point estimates and variability [62].

**Note 2:** For a detailed explanation of each item on the PEDro scale, please refer to the corresponding article [54].

* The maximum achievable score was 10 out of 10 possible points.

Furthermore, we conducted an Egger's test to examine the presence of publication bias. Sensitivity analyses were performed by systematically excluding individual studies to identify the potential sources of heterogeneity. The robustness of the findings was assessed by excluding specific studies [69].

## Results

### Selection of studies

The data depicted in Fig 1 reveal that 140 scholarly articles were initially gathered from the four databases and a further 9560 articles using Google Scholar. After manually eliminating duplicate entries and irrelevant records, we retained 623 articles based on title and abstract evaluations for further screening. Consequently, we identified 390 potential candidates from these papers, which warranted a comprehensive analysis through full-text assessment. Nevertheless, following scrutiny of the complete texts against our inclusion criteria, only 13 publications fulfilled all requirements and were included in our meta-analysis.

### Population characteristics

The demographic profiles of the 13 studies are as follows:

- Types of athletes: Three categories of male team-sport athletes relevant to this study were obtained from the 13 selected studies: four studies investigated basketball players [41, 44, 46, 49], seven investigated soccer players [42, 43, 45, 48, 51–53], and two investigated handball players [47, 50].

- Sample size: 13 studies included 286 players. The players' ages varied from 14 to 26, with median and mean ages of 18.2 and 18.8, respectively.

- Competitive level: Ten studies focused on athletes at the national level [43–45, 47–53], two focused on players at the regional or academic level [41, 42], and one on athletes at the college level [46].

- Sports experience: Of the 13 articles analyzed, only one did not investigate sports experience [49]. The athletes had different levels of campaign experience ranging from 2 to 18 years. The median and mean durations were 5.1, and 7 years, respectively.

### Intervention characteristics

The 13 studies exhibited the following attributes in terms of intervention characteristics:

- Training duration and session: The intervention period ranged from 5 to 9 weeks.

    Furthermore, intervention sessions ranged from 10 to 18, with the shortest and longest durations being 10 and 18 sessions, respectively.

- Training frequency: The training frequency was twice per week in 10 studies, three times per week in two studies, and 2–3 weeks with 15 training sessions in one study.

- Intervention type: All 13 studies compared pre- and post-test outcomes between experimental and control groups. Among these, eight conducted comparative analyses between HIIT and SSGs [41–45, 50–52], while the remaining studies compared HIIT with other types of exercise routines [46–49, 53].

- Intervention intensity: Nine articles reported training intensity encompassing specific parameters of content, number of repetitions, and sets per session [41, 42, 44, 46, 47, 50–53]. One study reported all-out training intensity without explicit parameters [48], whereas three studies did not provide any information regarding training intensity [43, 45, 49]. Among the nine articles, three demonstrated VIFT with an intensity range of 90–95% [41, 42, 44]. Four studies reported 90–95% maximal heart rate ($HR_{max}$) [46, 51–53]. Additionally, one study reported a 90–130% maximal aerobic speed (MAS) [47], and another utilized an individualized approach by applying 90% of the YYIRT1 final speed [50]. As most control group studies employed general routine training protocols, detailed specifications for training intensity indices were not provided. Only three studies incorporated training intensity indices, with two employing $HR_{max}$ indices of 85–95% [41, 45], whereas another study reported 85% MAS [52].

## Outcomes and meta-analysis

The results of this study were categorized according to the impact of HIIT on specific measures of physical fitness. The classification of physical fitness was examined across two primary domains: one encompassing measures of aerobic capacity, such as VO2max and YYIRT, and the other pertaining to RSA. Owing to limited data, as only two studies reported VIFT and one reported MAS, no meta-analysis was conducted; however, these findings will be discussed in detail later.

**Effect of HIIT on VO2max.** VO2max was evaluated in six studies [41–43, 46, 49, 53] comprising a total of six distinct sets, including both experimental and control groups (n = 150). To evaluate the influence of the HIIT intervention on aerobic capacity, specifically VO2max, a random-effects model was employed to compare the experimental and control groups. The results indicated a small ES of HIIT on VO2max (ES = 0.48; 95% [CIs] = 0.16–0.79; p = 0.003; $I^2$ = 0.0%; Egger's test p = 0.83; Fig 2). Each study contributed weight values ranging from 11.28% to 25.94% in the analysis.

**Effect of HIIT on YYIRT.** YYIRT was evaluated in six studies [41, 42, 45, 48, 50, 53] involving six experimental and control groups (n = 119). To evaluate the influence of the HIIT intervention on aerobic capacity measured by the YYIRT, a random-effects model was

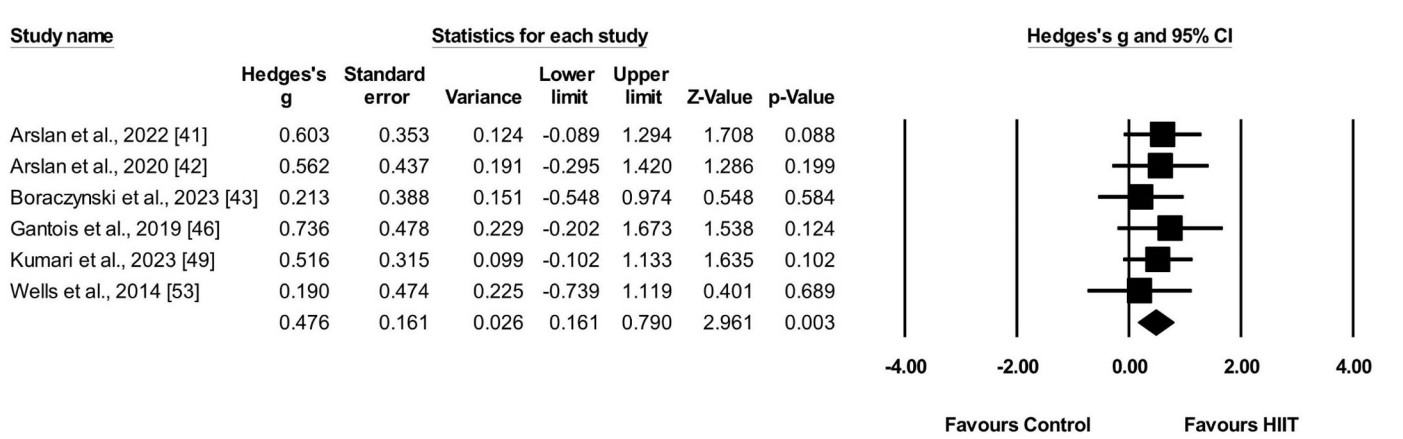

**Fig 2. Effect sizes of HIIT on VO2max in athletes with 95% confidence intervals (CIs).**

employed to compare the experimental and control groups. The findings revealed a moderate ES for HIIT in YYIRT (ES = 0.61, 95% CIs = 0.15–1.06; p = 0.009; $I^2$ = 35.4%). Egger's test showed a p-value of 0.28 (Fig 3). The weight values assigned to each study varied between 11.34% and 22.48%, and each study made a unique contribution to the analysis.

**Effect of HIIT on RSA.** In the present study, $RSA_{total}$, $RSA_{mean}$, and $RSA_{best}$ were used to assess repetitive sprinting ability. $RSA_{total}$ was evaluated by measuring the total time taken for six consecutive round-trip sprints of 20 m. A combined meta-analysis was conducted because of the similarity in ESs between $RSA_{mean}$ and $RSA_{best}$.

$RSA_{total}$ was evaluated in five studies [41, 42, 46, 47, 52], encompassing five experimental and control groups (n = 132). To evaluate the impact of the HIIT intervention on anaerobic performance in relation to $RSA_{total}$, a random-effects model was employed to compare the experimental and control groups. The study outcomes demonstrated a moderate ES of HIIT on $RSA_{total}$ (ES = 0.84; 95% CIs = 0.39–1.29; p < 0.001; $I^2$ = 34.3%; Egger's test p = 0.32; Fig 4). Each study contributed weight values ranging from 14.73% to 28.67% in the analysis.

$RSA_{mean}$ and $RSA_{best}$ were evaluated in four studies [45–47, 52], encompassing seven experimental and control groups (n = 102). To evaluate the effects of the HIIT intervention on anaerobic capacity, specifically $RSA_{mean}$ and $RSA_{best}$, a random-effects model was employed to compare the experimental and control groups. The results showed a moderate ES for HIIT on $RSA_{mean}$ and $RSA_{best}$ (ES = 0.89; 95% CIs = 0.58–1.19; p < 0.001; $I^2$ = 0.0%; Egger's test p = 0.90; Fig 5). Each study contributed weight values ranging from 11.61% to 17.47% in the analysis.

## Discussion

This study assessed the critical effect of HIIT on selected physical fitness parameters, including VO2max, YYIRT, RSA, VIFT, and MAS, in male team-sport athletes. It provides a theoretical framework for enhancing physical performance among male team-sport athletes and offers a comprehensive understanding of the HIIT content and methodologies. This study stands out from previous research because it specifically focuses on HIIT interventions in male team-sport athletes. The findings demonstrated significant improvements in VO2max, YYIRT, and RSA after HIIT; however, no statistically significant effects were observed for VIFT in the two relevant studies (ES = 0.20; p = 0.477). Moreover, the MAS effect was not significantly different between the HIIT and control groups (ES = 0.41; p = 0.403).

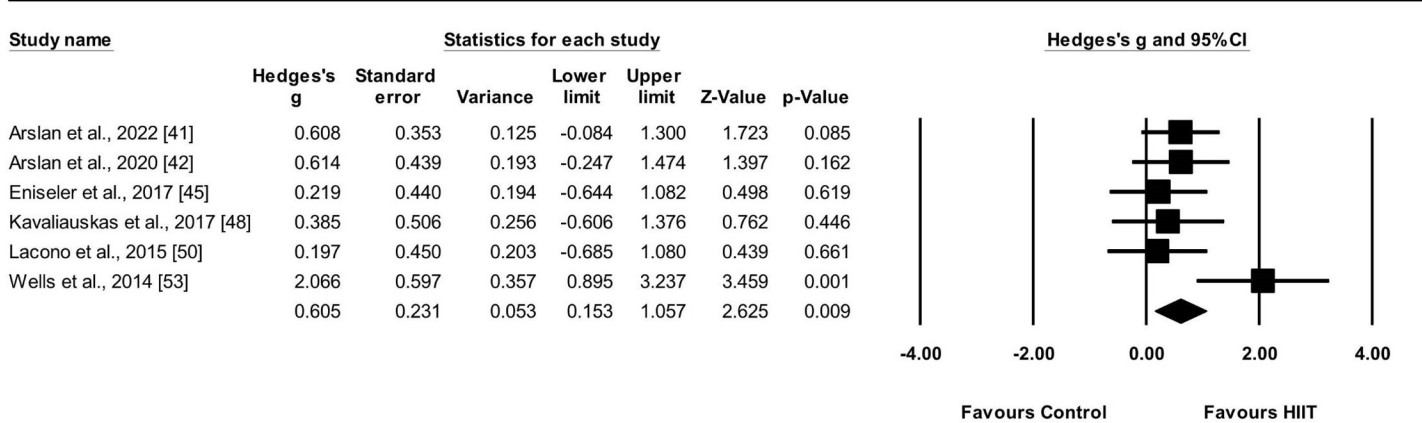

**Fig 3. Effect sizes of HIIT on YYIRT in athletes with 95% confidence intervals (CIs).**

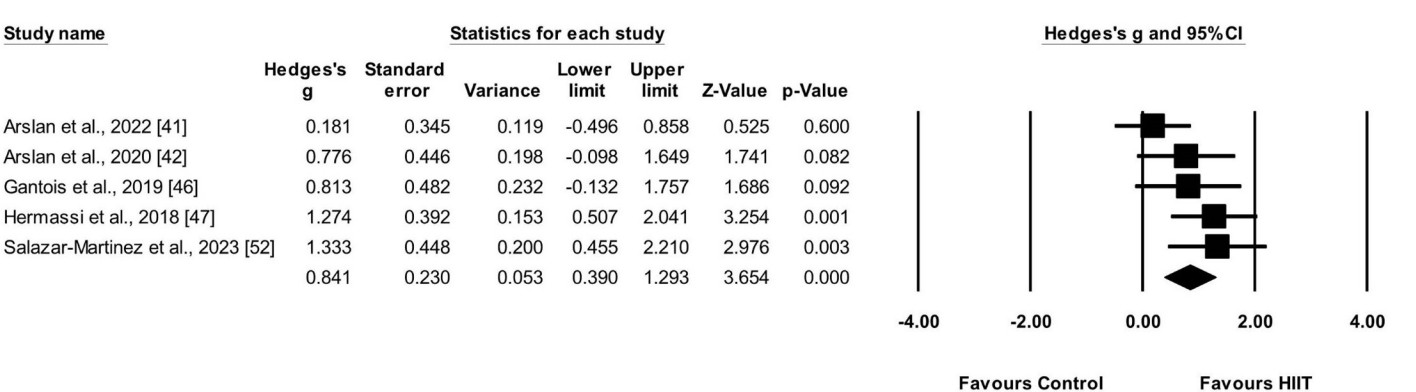

**Fig 4. Effect sizes of HIIT on RSA$_{total}$ in athletes with 95% confidence intervals (CIs).**

## Effect of HIIT on VO2max

The combined results of the six studies demonstrated a positive effect of HIIT on VO2max [41–43, 46, 49, 53]. However, the ES was small (0.48), indicating that the difference between the experimental and control groups was small, although statistically significant (p = 0.003). The results obtained align closely with those documented in previous research [16, 70, 71].

Regarding the type of sports program, six studies evaluated VO2max. Three studies evaluated soccer players [42, 43, 53], one of which reported no improvement [43], whereas the other two reported improvement [42, 53], with one showing significant improvement [42]. Furthermore, three studies evaluated basketball players [41, 46, 49] and reported improved results, one of which was significant [41]. Additionally, the findings indicated that HIIT had a minimal impact on soccer athletes compared with its effect on the control group (ES = 0.33). In contrast, HIIT had a moderate effect (ES = 0.61) on basketball players. In conclusion, the results showed that using HIIT training methods in the experimental group led to an increase

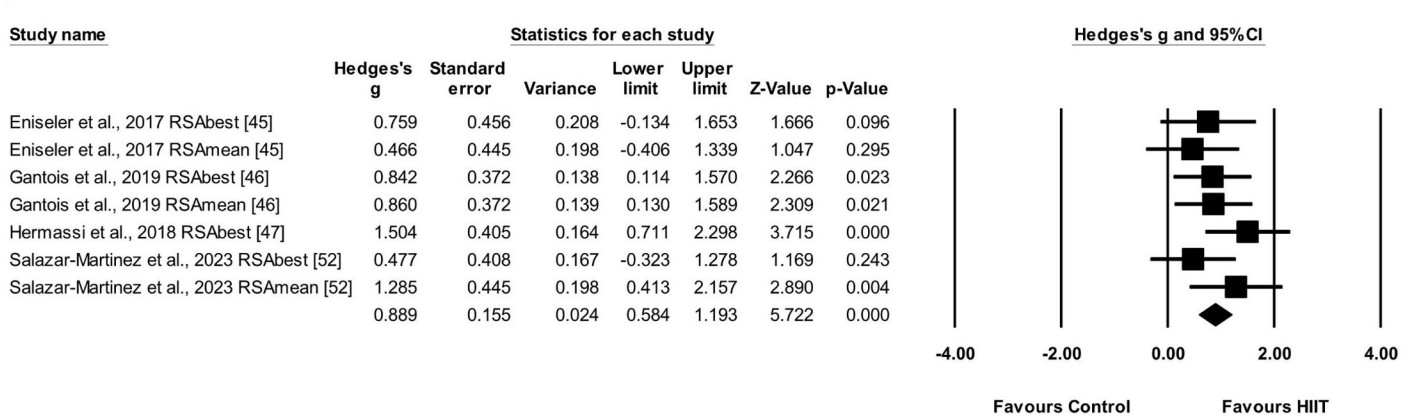

**Fig 5. Effect sizes of HIIT on RSA$_{mean}$ and RSA$_{best}$ in athletes with 95% confidence intervals (CIs).**

in VO2max compared with using a routine training program in the control group. This indicates that HIIT is more advantageous than traditional training.

Three studies conducted a comparative analysis between HIIT and conventional continuous training, demonstrating that HIIT exhibited greater practicality compared to continuous training. Additionally, three other studies examined and compared the effects of HIIT and SSG, indicating that both interventions yielded almost identical results without significant differences. Overall, sustained HIIT interventions were associated with an increased VO2max. The heterogeneity among the six included studies was zero, demonstrating consistency in the intervention effects of HIIT on VO2max, as revealed by the meta-analysis.

## Effect of HIIT on YYIRT

Five studies evaluated YYIRT1 [41, 42, 45, 48, 50] and one assessed YYIRT2 [53]. A notable enhancement was observed in YYIRT scores among all individuals (ES = 0.61; p = 0.009). Regarding the types of sports programs examined, one study focused on basketball players [41], four on soccer players [42, 45, 48, 53], and one on handball players [50]. These exercises represent the demanding nature of the cardiovascular endurance required in team sports, highlighting the significance of HIIT in enhancing cardiovascular endurance in athletes. The meta-analysis revealed low heterogeneity ($I^2$ = 35.5%), which was eliminated upon excluding a single study [53] that measured YYIRT2, whereas all other studies measured YYIRT1, resulting in inconsistent results. Nevertheless, the overall heterogeneity was minimal. In conclusion, the findings from these six studies consistently indicated the positive effects of HIIT in support of the experimental group compared to the control group.

## Effect of HIIT on RSA

RSA was assessed in six studies that demonstrated significant enhancements in composite indicators [41, 42, 45–47, 52]. Five studies assessed $RSA_{total}$ and reported a moderate ES (ES = 0.84) [41, 42, 46, 47, 52]. Sensitivity analyses revealed a small degree of heterogeneity ($I^2$ = 34.8%; p < 0.001), indicating consistency in the findings. Four studies analyzed $RSA_{best}$ and $RSA_{mean}$, demonstrating a moderate ES (ES = 0.89) and strong consistency in the results ($I^2$ = 0.0%, p < 0.001) [45–47, 52]. In summary, the implementation of HIIT demonstrated a notable enhancement in RSA among male athletes participating in team sports with a moderate effect.

With respect to the type of sports program, RSA was evaluated in six studies: one focused on basketball programs, one examined handball programs, and three investigated soccer programs. The use of HIIT is more prevalent among football players than among handball and basketball players for enhancing repetitive sprinting capacity. Generally, the interventional effect of HIIT in the experimental groups exceeded that in the control groups, indicating a positive effect of HIIT on RSA. The observed heterogeneity was minimal, suggesting a certain degree of consistency in the results.

## Effect of HIIT on VIFT

Two studies assessed the VIFT, which revealed a small ES (ES = 0.2). The investigations yielded no significant differences between the two groups (p = 0.443 and 0.870, respectively). However, caution should be exercised when generalizing these findings because of the limited number of studies.

## Limitations

This study comprehensively summarized and analyzed the effects of HIIT on VO2max, YYIRT, RSA, MAS, and VIFT in male team-sport athletes across various athletic profiles. Nevertheless, it is crucial to recognize the inherent constraints of this study, which include the following.

- Uneven age distribution: The variation in age across the studies (14–18 vs. 19–26 years) introduces a potential confounding factor. Physiological responses to HIIT can differ significantly between adolescents and young adults, making it difficult to draw definitive conclusions about their impact on specific age groups.

- Variation in athletic proficiency: Including athletes at national, regional, academic, and college levels creates a heterogeneous population. This variability in training experience and skill level can influence the results and limit the generalizability of the findings to specific athlete populations.

- Sample size concerns: Sample size selection is influenced by various factors, such as research objectives, population characteristics, potential risks associated with selecting an inadequate sample, and minimal sampling error [72]. Hence, the quality, credibility, and accuracy of the findings would be greatly affected if the sample sizes were deemed unsuitable, insufficient, or overly abundant [73]. The lack of clarity regarding sample size determination raises questions about the statistical power of the studies. Additionally, the relatively small sample sizes (average of 22.5 participants) increased the risk of sampling error and limited the ability to detect significant effects.

- Diverse control group interventions: The use of different control group interventions (SSG, routine training, and skill drills) introduced variability in group comparisons. This makes it challenging to isolate the specific effects of HIIT and compare results across studies.

- Variation in training duration: The varying durations of interventions (5–6 weeks vs. 7–9 weeks) added complexity to the analysis and interpretation of findings. The duration of HIIT programs can significantly influence their effect on physical performance, making it difficult to assess the effectiveness of specific protocols.

## Conclusions

This study comprehensively analyzed the effects of HIIT on male team sports athletes across diverse age groups, athletic levels, and disciplines. Our findings support the use of HIIT to improve several aspects of physical fitness, particularly VO2max, YYIRT, and RSA. This suggests that HIIT can enhance oxygen consumption performance, which is crucial in many male team sports. The experimental groups consistently outperformed the control groups in terms of VO2max, YYIRT, and RSA, demonstrating the efficacy of HIIT interventions. The lack of positive effects on the VIFT and MAS highlights the necessity for further research to explore how HIIT influences these specific aspects of physical fitness. The acknowledgment of the variability in results due to factors such as sports discipline, athlete profile, and intervention duration emphasizes the need for individualized HIIT program design.

## Supporting information

**S1 Table. Detailed search strategy.**
(DOCX)

**S2 Table. PRISMA 2020 checklist.**
(DOCX)

**S3 Table. Date used for meta-analysis.**
(DOCX)

**S4 Table. Data extraction table.**
(DOCX)

**S1 File. Literature search process.**
(ZIP)

## Author Contributions

**Conceptualization:** Yandong Yuan, Kim Geok Soh, Fengmeng Qi, Marrium Bashir.

**Data curation:** Yandong Yuan, Fengmeng Qi, Marrium Bashir.

**Formal analysis:** Marrium Bashir.

**Investigation:** Yandong Yuan.

**Methodology:** Yandong Yuan, Kim Geok Soh.

**Project administration:** Kim Geok Soh.

**Supervision:** Kim Geok Soh.

**Validation:** Fengmeng Qi, Marrium Bashir.

**Writing – original draft:** Yandong Yuan.

**Writing – review & editing:** Kim Geok Soh, Ningxia Zhao.

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
