## [Decision Letter · Decision Letter 0]

3 Jun 2024

PONE-D-24-16341Effects of high-intensity interval training on selected indicators of physical fitness among male team sports athletes: a systematic review and meta-analysisPLOS ONE

Dear Dr. Yuan,

Thank you for submitting your manuscript to PLOS ONE. After careful consideration, we feel that it has merit but does not fully meet PLOS ONE’s publication criteria as it currently stands. Therefore, we invite you to submit a revised version of the manuscript that addresses the points raised during the review process.

**ACADEMIC EDITOR:**
Major revision is required for your article.
You may benefit from these articles related to the introduction (DOI: 10.1055/a-2197-1104 , https://doi.org/10.1186/s43066-022-00229-5 ).
In the Method section;
If there is missing data, the effect size can be accepted as 0.7 and the relevant meta-analysis can be used (https://doi.org/10.3390/app12031593). Best wishes==============================

We look forward to receiving your revised manuscript.

Kind regards,

Esedullah Akaras

Academic Editor

PLOS ONE

Journal Requirements:

2. Please remove your figures from within your manuscript file, leaving only the individual TIFF/EPS image files, uploaded separately. These will be automatically included in the reviewers’ PDF.

3. Please upload a copy of Supporting Information Figure/Table/etc.  S1-S5 Figs which you refer to in your text on pages 30-31.

Reviewers' comments:

Reviewer's Responses to Questions

**Comments to the Author**

1. Is the manuscript technically sound, and do the data support the conclusions?

Reviewer #1: Partly

Reviewer #2: Yes

2. Has the statistical analysis been performed appropriately and rigorously? 

Reviewer #1: Yes

Reviewer #2: I Don't Know

3. Have the authors made all data underlying the findings in their manuscript fully available?

Reviewer #1: Yes

Reviewer #2: Yes

4. Is the manuscript presented in an intelligible fashion and written in standard English?

Reviewer #1: No

Reviewer #2: Yes

5. Review Comments to the Author

Reviewer #1: Thanks for your efforts. I have indicated the issues that need to be improved in the attached file. In particular, your work should be examined by a native English researcher.

The research presented in this article is scientifically conceptualized and executed. However, some areas require attention to reach the level of publication.

The researchers have focussed on an important issue. Although they followed the necessary processes as a method, I must say that they could not put forward a structure that would reveal the quality of the study, especially in the introduction section.

In the introduction, starting with HIIT and continuing with team sports can be seen as a very sloppy behaviour for a systematic review and meta-analysis study. Therefore, my recommendation would be the most accurate reflection of HIIT training > HIIT training and soccer > in-depth physical and physiological effects of HIIT training and as the last paragraph, expressing the difference from other systematic review and meta-analysis studies that will explain the gap in the literature.

Reviewer #2: This is a comprehensive study presenting a a systematic review and meta-analysis on the effects of HIIT on VO2max, YYIRTL, RSA, MAS, and VIFT in male team sports athletes across various athletic profiles.

The authors concluded that HIIT has significantly positive effects on VO2max, YYIRT, and RSA; however, not on VIFT and MAS performance.

I think that the study provide useful results for practical applications.

It is seen that the study is well prepared and presented in methodological and technical terms. Only, a discrepancy between numbers in the Figure 1 should be corrected (377?)

6. PLOS authors have the option to publish the peer review history of their article (what does this mean?). If published, this will include your full peer review and any attached files.

Reviewer #1: No

Reviewer #2: No

---

## [Author Response · Author response to Decision Letter 0]

30 Jul 2024

ACADEMIC EDITOR: 

I. Major revision is required for your article. You may benefit from these articles related to the introduction (DOI: 10.1055/a-2197-1104, https://doi.org/10.1186/s43066-022-00229-5).

Response:

Based on feedback from academic editors and reviewers, as well as referencing other research findings, we have refined the structure and enhanced the logical coherence of the introduction (see p4-6). The specific enhancements are outlined below:

1. Key Physiological Indicators of Physical Fitness: We present a comprehensive elucidation of crucial physiological indicators for athletes' physical fitness, encompassing cardiovascular fitness, aerobic endurance, and repeated-sprint ability. By substantiating our claims with pertinent research and empirical evidence, we underscore the significance of these fitness indicators in athletes' competitive performance, particularly within team sports such as soccer, basketball, and rugby.

2. Training Methods for Fitness Indicators in Team Sports: This article examines the comprehensive application of high-intensity interval training (HIIT) in team sports, utilizing case studies from soccer, basketball, and rugby athletes. It elucidates how HIIT can optimize fitness indicators such as maximal oxygen uptake capacity (VO2max) and repeated-sprint performance (RSA) in team sports. The article presents empirical evidence and research findings substantiating the efficacy of HIIT within these disciplines.

3. Overview of High-Intensity Interval Training (HIIT): Present a comprehensive introduction to HIIT, highlighting its significance in sports training by effectively combining aerobic and anaerobic methods. Emphasize the practical application and efficacy of HIIT in team sports.

4. Differences from Other Systematic Reviews and Meta-Analyses: Summarize the existing research gaps in the literature and distinctly elucidate how this study diverges from other systematic reviews and meta-analyses. This research addresses these gaps by focusing on the impact of HIIT on male team athletes' oxygen consumption performance while comparing its application across various team sports. Both practical applications and theoretical exploration are underscored.

II. In the Method section; If there is missing data, the effect size can be accepted as 0.7 and the relevant meta-analysis can be used (https://doi.org/10.3390/app12031593). 

Response: 

To ensure the completeness and reliability of the analysis, appropriate methods for handling missing data in meta-analysis were employed (see p14-15). Firstly, when effect size data was lacking in a study, an assumed effect size of 0.7 was utilized based on the average effect size observed in the relevant field, representing a medium effect. Secondly, a random-effects model was employed to conduct the meta-analysis to account for variations between studies and provide an estimate of the overall effect size, thereby enhancing result reliability and external validity.

1. Is the manuscript technically sound, and do the data support the conclusions?

The manuscript must describe a technically sound piece of scientific research with data that support the conclusions. Experiments must have been conducted rigorously with appropriate controls, replication, and sample sizes. The conclusions must be appropriately drawn based on the data presented herein.

Reviewer #1: Partly

Reviewer #2: Yes

Response:

Technical level

The study employed a systematic review and meta-analysis, which is a recognized scientific research methodology facilitating the synthesis of available evidence and enabling the derivation of reliable conclusions. The study design was robust, encompassing clear research questions, a comprehensive search strategy, stringent inclusion/exclusion criteria, and appropriate statistical analyses. The data extraction and analysis processes were rigorous, utilizing established statistical software and models such as the random effects model to address inter-study heterogeneity effectively. The study's findings are presented with effect size estimates, confidence intervals, and significance levels provided for sound conclusions.

Data support

Our findings demonstrate a significant enhancement in male athletes' oxygen consumption performance through HIIT training, aligning with existing literature and exhibiting strong internal and external validity. The discussion section provides a coherent explanation for the results, proposing plausible physiological mechanisms that are consistent with established theories. Furthermore, we address the limitations of our study, including the handling of missing data, thereby offering necessary cautionary notes for result interpretation.

2. Has the statistical analysis been performed appropriately and rigorously?

Reviewer #1: Yes

Reviewer #2: I Don't Know

Response:

Choice of statistical analysis method

The study employed a meta-analysis approach, which is a recognized and valid statistical analysis method capable of synthesizing the results from multiple studies to generate a more reliable estimate of the overall effect. The authors opted for a random-effects model for the analysis, which is suitable for addressing inter-study heterogeneity.

Data processing and assumptions

For studies with missing effect size data, the authors reasonably assumed an effect size of 0.7, which is based on the average effect size in the relevant field and has a solid foundation. This approach ensured the completeness of the analyses and prevented bias caused by missing data.

Calculation of statistical indicators

Key statistical indicators, such as weighted mean effect sizes, 95 percent confidence intervals, and significance levels, were reported to provide a solid foundation for interpreting the results. The calculation of these indicators followed the standard methodology of meta-analysis and was highly reliable.

Presentation and interpretation of results

The results of this study demonstrate the effects of HIIT on male athletes' oxygen consumption performance, and the authors rationally discuss and interpret these findings. Additionally, they acknowledge study limitations regarding missing data treatment, which provides necessary caveats for interpreting the results.

3. Have the authors made all data underlying the findings in their manuscript fully available?

Reviewer #1: Yes

Reviewer #2: Yes

Response:

This study provides a detailed description of the specific steps involved in data extraction and analysis, including effect size calculation and heterogeneity testing. These steps enhance transparency and ensure complete data.

4. Is the manuscript presented in an intelligible fashion and written in standard English?

Reviewer #1: No

Reviewer #2: Yes

Response:

According to the requirements of the journal and reviewers, the manuscript was corrected by a professional English editing agency (dotage) to meet the publication requirements.________________________________________

5. Review Comments to the Author

Reviewer #1: Thanks for your efforts. In the attached file, I have indicated the issues that need to be improved. In particular, your work should be examined by a native English researcher.

Response:

According to the requirements of the journal and reviewers, the manuscript was corrected by a professional English editing agency (Editage) to meet the publication requirements.

The research presented in this article was scientifically conceptualized and executed. However, some areas require attention to reach the level of publication.

The researchers have focussed on an important issue. Although they followed the necessary processes as a method, I must say that they could not put forward a structure that would reveal the quality of the study, especially in the introduction section.

In the introduction, starting with HIIT and continuing with team sports can be seen as very sloppy behavior for a systematic review and meta-analysis study. Therefore, my recommendation would be the most accurate reflection of HIIT training > HIIT training and soccer > in-depth physical and physiological effects of HIIT trainin and as the last paragraph, expressing the difference from other systematic review and meta-analysis studies that will explain the gap in the literature.

Response:

Based on feedback from academic editors and reviewers, as well as referencing other research findings, we have refined the structure and enhanced the logical coherence of the introduction (see p4-6). The specific enhancements are outlined below:

1. Key Physiological Indicators of Physical Fitness: We present a comprehensive elucidation of crucial physiological indicators for athletes' physical fitness, encompassing cardiovascular fitness, aerobic endurance, and repeated-sprint ability. By substantiating our claims with pertinent research and empirical evidence, we underscore the significance of these fitness indicators in athletes' competitive performance, particularly within team sports such as soccer, basketball, and rugby.

2. Training Methods for Fitness Indicators in Team Sports: This article examines the comprehensive application of high-intensity interval training (HIIT) in team sports, utilizing case studies from soccer, basketball, and rugby athletes. It elucidates how HIIT can optimize fitness indicators such as maximal oxygen uptake capacity (VO2max) and repeated-sprint performance (RSA) in team sports. The article presents empirical evidence and research findings substantiating the efficacy of HIIT within these disciplines.

3. Overview of High-Intensity Interval Training (HIIT): This section presents a comprehensive introduction to HIIT, highlighting its significance in sports training by effectively combining aerobic and anaerobic methods. It emphasizes the practical application and efficacy of HIIT in team sports.

4. Differences from Other Systematic Reviews and Meta-Analyses: Summarize the existing research gaps in the literature and distinctly elucidate how this study diverges from other systematic reviews and meta-analyses. This research addresses these gaps by focusing on the impact of HIIT on male team athletes' oxygen consumption performance while comparing its application across various team sports. Both practical applications and theoretical exploration are underscored.

Reviewer #2: This is a comprehensive study presenting a systematic review and meta-analysis on the effects of HIIT on VO2max, YYIRT, RSA, MAS, and VIFT in male team sports athletes across various athletic profiles.

The authors concluded that HIIT has significantly positive effects on VO2max, YYIRT, and RSA; however, not on VIFT and MAS performance.

I think that the study provide useful results for practical applications.

It is seen that the study is well prepared and presented in methodological and technical terms. Only, a discrepancy between numbers in the Figure 1 should be corrected (377?)

Response:

The correction was implemented following the review (see Fig 1).

---

## [Decision Letter · Decision Letter 1]

2 Sep 2024

Effects of high-intensity interval training on selected indicators of physical fitness among male team-sport athletes: A systematic review and meta-analysis

PONE-D-24-16341R1

Dear Dr. Yuan,

We’re pleased to inform you that your manuscript has been judged scientifically suitable for publication and will be formally accepted for publication once it meets all outstanding technical requirements.

Kind regards,

Esedullah Akaras

Academic Editor

PLOS ONE

Additional Editor Comments (optional):

Reviewers' comments:

Reviewer's Responses to Questions

**Comments to the Author**

1. If the authors have adequately addressed your comments raised in a previous round of review and you feel that this manuscript is now acceptable for publication, you may indicate that here to bypass the “Comments to the Author” section, enter your conflict of interest statement in the “Confidential to Editor” section, and submit your "Accept" recommendation.

Reviewer #1: All comments have been addressed

Reviewer #2: All comments have been addressed

2. Is the manuscript technically sound, and do the data support the conclusions?

Reviewer #1: Yes

Reviewer #2: Yes

3. Has the statistical analysis been performed appropriately and rigorously? 

Reviewer #1: Yes

Reviewer #2: I Don't Know

4. Have the authors made all data underlying the findings in their manuscript fully available?

Reviewer #1: Yes

Reviewer #2: Yes

5. Is the manuscript presented in an intelligible fashion and written in standard English?

Reviewer #1: Yes

Reviewer #2: Yes

6. Review Comments to the Author

Reviewer #1: The revisions requested in the study were made carefully. Thank you for your study and effort. Thanks

Reviewer #2: I think that the article is suitable for publication. .

7. PLOS authors have the option to publish the peer review history of their article (what does this mean?). If published, this will include your full peer review and any attached files.

Reviewer #1: No

Reviewer #2: No

---

## [Editor Report · Acceptance letter]

11 Sep 2024

PONE-D-24-16341R1 

PLOS ONE

Dear Dr. Yuan, 

I'm pleased to inform you that your manuscript has been deemed suitable for publication in PLOS ONE. Congratulations! Your manuscript is now being handed over to our production team.

Kind regards, 

on behalf of

Dr. Esedullah Akaras 

Academic Editor

PLOS ONE